# Long-Term Application of Organic Fertilizers in Relation to Soil Organic Matter Quality

Ondřej Sedlář [1,*], Jiří Balík [1], Jindřich Černý [1], Martin Kulhánek [1] and Michaela Smatanová [2]

1   Department of Agro-Environmental Chemistry and Plant Nutrition, Faculty of Agrobiology, Food and Natural Resources, Czech University of Life Sciences Prague, 16500 Praha, Czech Republic
2   Department of Plant Nutrition, Central Institute for Supervising and Testing in Agriculture, 60300 Brno, Czech Republic
*   Correspondence: sedlar@af.czu.cz; Tel.: +420-22438-2743

**Abstract:** The quality of soil organic matter plays a central role in soil structure, carbon sequestration and pollutant immobilization. The effect of 16–23 years of fertilization on the quality of soil organic matter was studied in field experiments at ten experimental sites in Central Europe. Soil samples were collected in 2016 after barley harvest. Six crops were rotated: pea–canola–winter wheat–spring barley–beet/potato–spring barley. Six treatments were studied: unfertilized control, mineral fertilization (NPK), farmyard manure, farmyard manure + NPK, straw incorporation, and straw incorporation + NPK. Although carbon input did not significantly correlate with any soil organic carbon fractions, the C/N ratio of applied organic fertilizers significantly correlated with the content of humic acid carbon (C-HA), the C-HA/C-FA ratio and humification index in soil. The combination of farmyard manure + NPK resulted in a higher humic acid carbon content in soil, humification rate, and humification index compared to the application of NPK, straw return, and the combination of straw return + NPK. Although straw return led to a lower E4/E6 (A400/A600, Q4/6) ratio compared to farmyard manure application, the C-HA/C-FA ratio was unchanged among these treatments. The application of farmyard manure with and without the addition of NPK led to higher values of carbon sequestration efficiency in soil compared to the straw return with and without the addition of NPK.

**Keywords:** fulvic acid; HA/FA; humic acid; humification; sequestration

## 1. Introduction

Additions of organic manures result in increased soil organic matter content. Many reports have shown that this results in increased water holding capacity, porosity, infiltration capacity, hydraulic conductivity and water stable aggregation and decreased bulk density and surface crusting [1]. A straw return is the main method of crop straw treatment. However, the straw return method commonly used has many adverse effects on the levels and improvement of soil fertility and crop yield [2]. The application of mineral fertilizers also results in an increase in the amount of organic matter returned to the soil [1].

There are two means to increase the organic matter content in soils; one is to increase the organic matter gains or additions to the soil, and the other is to decrease organic matter losses. Storage of soil organic carbon is a balance between carbon additions from non-harvested portions of crops and organic amendments and carbon losses, primarily through organic matter decomposition and release of respired $CO_2$ to the atmosphere. Organic matter returned to the soil, directly from crop residues or indirectly as manure, consists of many different organic compounds. Some of these are digested quickly by soil microorganisms. The result of this is a rapid formation of microbial compounds and body structures, which are important in holding particles together to provide soil structure and limit soil erosion, and the release of carbon dioxide back to the atmosphere through

microbial respiration. Thus, soil organic matter is not only an important source of carbon for soil processes but also a sink for carbon sequestration. Cultivation can reduce soil organic carbon content and lead to soil deterioration, and finally reduce soil productivity [3].

The favorable quality of the humus positively influences the stabilization of water-stable soil aggregates, which are the basic units of the soil structure. As a consequence of soil cultivation, macroaggregates are broken into microaggregates which are stabilized mostly by humic acid carbon with highly condensed and stabilized macromolecules [4].

Humic substances are a major stable part of soil organic carbon that play a central role in soil carbon accumulation [5]. Humification depends on soil organic matter contents [6] and the C/N ratio of a particular fertilizer [7]. The continuous application of farmyard manure to field soils generally results in a higher humic acid content compared to the application of mineral fertilizers alone [8]. Furthermore, a regular application of rotted or composted farmyard manure within the rotation can increase soil organic carbon content much more than the separate application of straw and cattle slurry [5].

Both humic and fulvic acids have high sorption capacity with respect to many contaminants, including heavy metals, which can result in their immobilization and consequently protection of food and groundwater against contamination [9]. The increase in water-soluble sugars, "non-matured" lignin and fulvic acids is an indicator of a labile system characterized by a rapid course of changes and a longer period of stabilization or acquisition of dynamic equilibrium in the mineralization and humification process [10]. Furthermore, the high humic acid to fulvic acid ratio may explain decreased concentrations of metals in plants [11]. Heclik et al. [12] describe this phenomenon in their study on nanoparticles; fulvic acid molecules only form a salt with heavy metal ions, while the conformation of humic acid molecules is responsible for metal ion capture.

The E4/E6 ratio is inversely related to the degree of condensation of the aromatic network in humic acids. A low E4/E6 ratio is indicative of a relatively high degree of aromatic constituent condensation while a high ratio reflects a low degree of aromatic condensation and the presence of relatively large proportions of aliphatic structures [13]. Therefore, the humic to fulvic acid ratio is increased together with decreasing values of the E4/E6 ratio measured in the visible spectrum range [14].

Reliable quantitative evaluation of humic substances formation using, for example, parameters such as the humification index, humification degree and humification rate requires data from long-term experiments which are lacking because they are usually costly and time-consuming [9]. Therefore, this study aims to evaluate the long-term application of mineral fertilizers, farmyard manure and plant residue incorporation on the quality of soil organic matter.

## 2. Materials and Methods

Long-term on-farm trials have been established within the years 1993 and 2000 by the Central Institute for Supervising and Testing in Agriculture at ten experimental sites with various soil-climatic conditions in the Czech Republic. The characteristics of the experimental sites are given in Table 1. The average total organic carbon content varied between 8.1 and 15.0 g/kg. Within these trials, six crops were rotated in the following order: pea, canola, winter wheat, spring barley (1), beet/potato, and spring barley (2).

**Table 1.** Characteristics of the experimental sites and year of establishment of the experiment.

| Location | Root Crop | Since | Altitude (m) | Precip. [1] (mm/year) | Air Temp. [1] (°C) | Soil Group | $C_{HS}$ (%) | TOC (%) |
|---|---|---|---|---|---|---|---|---|
| 1. Horažd'ovice | potato | 1994 | 472 | 585 | 7.8 | Cambisol | 0.405 ± 0.061 | 1.34 ± 0.02 |
| 2. Hradec n. S. | potato | 1993 | 460 | 616 | 7.4 | Haplic Luvisol | 0.286 ± 0.042 | 1.19 ± 0.05 |
| 3. Chrastava | potato | 2000 | 345 | 738 | 8 | Haplic Luvisol | 0.421 ± 0.085 | 1.08 ± 0.04 |
| 4. Jaroměřice | potato | 1994 | 425 | 488 | 8.2 | Haplic Luvisol | 0.552 ± 0.102 | 1.26 ± 0.06 |
| 5. Lípa | potato | 1993 | 505 | 594 | 7.5 | Cambisol | 0.545 ± 0.095 | 1.35 ± 0.06 |
| 6. Lednice | beet | 1994 | 172 | 461 | 9.6 | Chernozem | 0.368 ± 0.102 | 1.57 ± 0.03 |
| 7. Pusté Jakartice | beet | 1994 | 290 | 584 | 8.3 | Retisol | 0.445 ± 0.114 | 0.88 ± 0.01 |
| 8. Staňkov | potato | 1994 | 370 | 549 | 8.3 | Haplic Luvisol | 0.470 ± 0.070 | 1.05 ± 0.04 |
| 9. Věrovany | beet | 1993 | 207 | 502 | 8.7 | Chernozem | 0.502 ± 0.038 | 1.29 ± 0.04 |
| 10.Vysoká | potato | 2000 | 595 | 611 | 7.1 | Cambisol | 0.484 ± 0.066 | 1.48 ± 0.08 |

[1] long-term (30 years) annual average. $C_{HS}$—humic substances carbon in the year 2020, TOC—total organic carbon in soil during the years 2011–2018.

As follows from Table 2, six treatments were studied: unfertilized treatment (unfert.), mineral fertilization (NPK)–basal application of $Ca(H_2PO_4)_2$ + KCl + $(NH_4)_2SO_4$ and top-dressing of calcium ammonium nitrate, application of farmyard manure (FYM), a combination of farmyard manure and mineral fertilization (FYM + NPK), incorporation of plant residues (STRAW/BT) and a combination of plant residues incorporation and mineral fertilization (STRAW/BT + NPK). The supply of nutrients in mineral fertilizers respected both the demands of crops and the maintenance of the optimal content ('good') of the available nutrients (Mehlich 3) nutrients in soil (K 170–310 mg/kg, P 80–115 mg/kg). At the sites of Lednice, Pusté Jakartice and Věrovany, beet was grown instead of potato, which resulted in the incorporation of the beet tops into the soil (Tables 1 and 2). The incorporation of cereal and canola straw was accompanied by the application of 40 kg N/ha and 20 kg N/ha, respectively. Each treatment had four replicates. Furthermore, each replicate was repeated three times during the soil analysis.

**Table 2.** Carbon input (t/ha) and C/N ratio of organic fertilizers applied in the individual treatments.

| Crop | Org. Fert. | Treatment | | | | | |
|---|---|---|---|---|---|---|---|
| | | Unfert. | NPK | FYM | FYM + NPK | STRAW/BT | STRAW/BT + NPK |
| pea | barley straw | | | | | 1.58 C/N 82 | 1.58 C/N 82 |
| canola | pea straw | | | | | 0.43 C/N 25 | 0.43 C/N 25 |
| | FYM | | | 2.10 C/N 30 | 2.10 C/N 30 | | |
| winter wheat | canola straw | 1.55 C/N 70 | 1.55 C/N 70 | 1.55 C/N 70 | 1.55 C/N 70 | 1.55 C/N 70 | 1.55 C/N 70 |
| spring barley | wheat straw | | | | | 1.58 C/N 82 | 1.58 C/N 82 |
| potato/ beet | barley straw | | | | | 1.58 C/N 82 | 1.58 C/N 82 |
| | FYM | | | 2.80 C/N 30 | 2.80 C/N 30 | | |

**Table 2.** *Cont.*

| Crop | Org. Fert. | Treatment | | | | | |
|---|---|---|---|---|---|---|---|
| | | Unfert. | NPK | FYM | FYM + NPK | STRAW/BT | STRAW/BT + NPK |
| spring barley | beet tops * | | | | | 1.89 C/N 16 | 1.89 C/N 16 |
| ∑ C input per rotation | | 1.55 | 1.55 | 6.45 | 6.45 | 6.73 (8.61 *) | 6.73 (8.61 *) |
| A weighted average of C/N in input per rotation | | 70.0 | 70.0 | 38.0 | 38.0 | 75.7 (60.7 *) | 75.7 (60.7 *) |

* if beet instead of potato is grown. FYM—farmyard manure. Fresh weight of applied organic fertilizers: 4 t/ha of cereal straw, 4 t/ha of canola straw, 1 t/ha of pea straw, 30 t/ha of beet tops, 30 t/ha and 40 t/ha of farmyard manure applied to canola and potato/beet, respectively.

Due to missing data related to the composition of organic fertilizers applied in all years of the experiments, the following parameters in dry matter were used to calculate carbon input and the C/N ratio: cereal straw 44% C [15,16] and a C/N ratio of 82 [15,17], pea straw 45% C [18,19] and a C/N ratio of 25 [20,21], beet tops 37% C [22,23] and a C/N ratio of 16 [21,23], farmyard manure 35% C [24,25] and a C/N ratio of 30 [26,27].

Soil samples were collected in 2016 after spring barley harvest (1) and analyzed as follows: A sample of 5.0 g of soil was stirred for 10 min in a mixture of 0.1 mol/L sodium pyrophosphate and 0.1 mol/L sodium hydroxide solution. After 24 h of storage, a saturated solution of sodium sulfate was added. A filtration followed. The filtrate formed was used for:

1. The E4/E6 ratio measurement directly in the filtrate. For the E4/E6 ratio, a visible light spectrometer Lambda 25 (PerkinElmer, Waltham, MA, USA) was used to calculate the specific spectral absorbance ratio at 465 and 665 nm [28].
2. Determination of carbon in humic substances. The filtrate was neutralized by sulphuric acid and then vaporized. Iodometric titration followed.
3. Determination of carbon in fulvic acids. The filtrate was acidified by sulphuric acid to a pH of 1.0–1.5 and warmed up for 30 min. After storage for 24 h, the solution was filtrated and washed using the 0.05 mol/L sulphuric acid solution. The newly formed filtrate was vaporized. Iodometric titration followed.

The carbon of humic acids was determined. The remaining precipitate was dissolved using the hot 0.05 mol/L sodium hydroxide solution. After neutralization by sulfuric acid and vaporization, iodometric titration followed.

Before titration of all samples, dry matter formed by vaporization was dissolved in a mixture of the 0.067 mol/L potassium dichromate solution and concentrated sulfuric acid and warmed up for 45 min.

Humification indices were calculated according to Raiesi [29] and Iqbal et al. [30]:

$$\text{degree of polymerization: } C_{HA}/C_{FA} \tag{1}$$

$$\text{humification rate: HR} = (C_{FA} + C_{HA})/\text{TOC} \tag{2}$$

$$\text{humification index: HI} = C_{HA}/\text{TOC} \tag{3}$$

where $C_{FA}$ is the fulvic acid carbon, $C_{HA}$ is the humic acid carbon and TOC is the total organic carbon in soil. The total organic carbon content in soil was determined from about 50 mg of soil by the modified Dumas combustion method at 960 °C with a CHNS Vario MACRO cube analyzer (Elementar, Langenselbold, Germany).

Except for the humification index, the studied variables significantly differed among experimental sites. Therefore, the effect of treatments was also evaluated by replacing the current values of variables with relative ones. Relative values were calculated as:

$$V_{\text{treatment}}/V_{\text{site-average}} \tag{4}$$

where $V_{treatment}$ was the value of each treatment, and $V_{site\text{-}average}$ was the average value of a particular site among all treatments.

Relative contribution (RC) of organic fertilizers to soil organic carbon stock was calculated according to Wang et al. [31]. The unfertilized treatment (0) and the NPK treatment, respectively, were taken to be the "control"; the FYM and STRAW/BT treatments were compared with the unfertilized treatment, whereas the FYM + NPK and STRAW/BT + NPK treatments were compared with the NPK treatment.

$$RC\ (\%) = [(TOC_{treatment} - TOC_{control})/TOC_{control}] \times 100 \qquad (5)$$

The carbon sequestration efficiency (CSE) was calculated as follows:

$$CSE\ (\%) = [(TOC_{treatment} - TOC_{control})/TCI] \times 100 \qquad (6)$$

where TCI is the total C input (t/ha) applied in organic fertilizers during the duration of individual experiments [31].

A one-way analysis of variance (ANOVA) using Fisher's LSD test was calculated. Pearson's correlation coefficients were used to analyze the relationships among the variables studied in Table 4. The probability value of 0.05 or less ($p < 0.05$) was considered statistically significant. A statistical analysis of the data was carried out using the Statistica version 13.3 software (TIBCO Software, Palo Alto, Santa Clara, CA, USA).

## 3. Results

As is shown in Table 3, unlike the content of fulvic acid carbon ($C_{FA}$), the content of humic acid carbon ($C_{HA}$) correlated significantly with the weighted average of the C/N ratio of applied organic fertilizers, RC and CSE (moderate correlations).

**Table 3.** Pearson's correlation coefficients (r) among variables.

| Var. | $C_{FA}$ | $C_{HA}$ | $C_{HA}/C_{FA}$ | E4/E6 | HR | HI | C input | C/N input | RC | CSE |
|---|---|---|---|---|---|---|---|---|---|---|
| $C_{HA}$ | 0.42 ** | | | | | | | | | |
| $C_{HA}/C_{FA}$ | −0.32 * | 0.51 *** | | | | | | | | |
| E4/E6 | 0.03 | −0.15 | −0.38 * | | | | | | | |
| HR | 0.73 *** | 0.61 *** | −0.09 | 0.03 | | | | | | |
| HI | 0.52 *** | 0.85 *** | 0.33 * | −0.11 | 0.87 *** | | | | | |
| C input | −0.15 | −0.14 | 0.24 | −0.48 ** | −0.04 | −0.04 | | | | |
| C/N input | 0.02 | −0.43 ** | −0.37 * | −0.03 | −0.23 | −0.45 ** | 0.32 * | | | |
| RC | 0.22 | 0.41 ** | 0.05 | 0.31 * | 0.19 | 0.32 * | −0.59 *** | −0.54 *** | | |
| CSE | 0.25 | 0.50 *** | 0.16 | 0.25 | 0.23 | 0.38 * | −0.47 ** | −0.50 ** | 0.96 *** | |
| precip. | −0.00 | −0.24 | −0.43 ** | 0.49 ** | 0.03 | −0.11 | −0.29 | 0.10 | 0.07 | 0.03 |
| temp. | −0.25 | 0.07 | 0.53 *** | −0.47 ** | −0.05 | 0.11 | 0.43 ** | −0.15 | −0.40 * | −0.37 * |

The r-values marked with asterisks are significant at the levels of significance * $p < 0.05$, ** $p < 0.01$, and *** $p < 0.001$. $C_{FA}$—fulvic acids carbon, $C_{HA}$—humic acids carbon, $C_{HA}/C_{FA}$—humic to fulvic acid carbon ratio, E4/E6—absorbances ratio at the wavelengths of 465 and 665 nm, HR—humification rate HR = $(C_{FA} + C_{HA})/TOC$, HI—humification index HI = $C_{HA}/TOC$, C input—carbon amount applied in organic fertilizers during the duration of individual experiments, C/N input—weighted average of C/N in organic fertilizers per rotation, RC–relative contribution of organic fertilizers to soil organic carbon stock, CSE–carbon sequestration efficiency, precip.—long-term annual precipitation, temp.—long-term average annual air temperature.

The E4/E6 ratio correlated significantly with the humic to fulvic acid carbon ($C_{HA}/C_{FA}$) ratio, although this correlation was weak. Although the E4/E6 ratio was significantly correlated with carbon input in organic fertilizers (moderate correlation), the $C_{HA}/C_{FA}$ ratio was significantly correlated with the C/N ratio of organic fertilizers (weak correlation). The long-term annual average of both precipitation amount and air temperature was moderately correlated with both the $C_{HA}/C_{FA}$ and the E4/E6 ratio. Higher values of the $C_{HA}/C_{FA}$ ratio and lower values of the E4/E6 ratio were recorded under conditions of

lower precipitation amount and higher air temperature. A stronger correlation was found between the $C_{HA}/C_{FA}$ ratio and the $C_{HA}$ content, rather than the $C_{FA}$ content.

Unlike the humification rate, the humification index correlated significantly with the weighted average of the C/N ratio of applied organic fertilizers (moderate correlation), RC (weak correlation) and CSE (weak correlation).

Both the relative contribution of organic fertilizers to soil organic carbon stock (RC) and carbon sequestration efficiency (CSE) correlated positively with the $C_{HA}$ content and negatively with both carbon input in organic fertilizers and the weighted average of the C/N ratio of applied organic fertilizers. All these correlations were moderate.

Carbon input did not correlate significantly with any fractions of soil organic carbon in soil. In contrast, the weighted average of the C/N ratio of applied organic fertilizers correlated significantly with the $C_{HA}$ content and the humification index (moderate correlation), and with the $C_{HA}/C_{FA}$ ratio (weak correlation).

### 3.1. Organic Carbon Fractions

The relative values of studied variables independent of the site effect are often mentioned in the following tables (Tables 4–6). The FYM + NPK treatment led to an increase in the relative $C_{FA}$ content compared to the NPK and FYM treatments (Table 4). Except for the FYM + NPK treatment, no significant differences in the $C_{FA}$ content among other treatments were found.

**Table 4.** Content of the fulvic acids carbon ($C_{FA}$) and humic acids carbon ($C_{HA}$) in soil at the individual experimental sites.

| Site/TRT | Unfert. | NPK | FYM | FYM + NPK | STRAW/BT | STRAW/BT + NPK |
|---|---|---|---|---|---|---|
| $C_{FA}$ (%) | | | | | | |
| 1 | 0.186 [b] | 0.270 [c] | 0.225 [a] | 0.231 [a] | 0.180 [b] | 0.240 [a] |
| 2 | 0.229 [b] | 0.156 [a] | 0.159 [a] | 0.157 [a] | 0.127 [a] | 0.166 [a] |
| 3 | 0.284 [c] | 0.197 [ab] | 0.213 [abc] | 0.242 [bc] | 0.151 [a] | 0.199 [ab] |
| 4 | 0.287 [a] | 0.240 [a] | 0.219 [a] | 0.256 [a] | 0.271 [a] | 0.242 [a] |
| 5 | 0.239 [a] | 0.244 [a] | 0.242 [a] | 0.381 [c] | 0.229 [a] | 0.308 [b] |
| 6 | 0.178 [b] | 0.153 [ab] | 0.136 [a] | 0.270 [c] | 0.135 [a] | 0.137 [a] |
| 7 | 0.198 [a] | 0.155 [c] | 0.224 [ab] | 0.199 [a] | 0.238 [b] | 0.203 [ab] |
| 8 | 0.248 [a] | 0.269 [a] | 0.229 [a] | 0.269 [a] | 0.295 [a] | 0.241 [a] |
| 9 | 0.197 [bc] | 0.119 [e] | 0.155 [a] | 0.170 [ab] | 0.251 [d] | 0.232 [cd] |
| 10 | 0.254 [ab] | 0.235 [ab] | 0.211 [a] | 0.237 [ab] | 0.267 [b] | 0.252 [ab] |
| relative $C_{FA}$ [1] | 1.065 [ab] | 0.924 [a] | 0.921 [a] | 1.103 [b] | 0.975 [ab] | 1.012 [ab] |
| $C_{HA}$ (%) | | | | | | |
| 1 | 0.146 [ab] | 0.177 [a] | 0.181 [a] | 0.191 [a] | 0.113 [b] | 0.147 [ab] |
| 2 | 0.124 [a] | 0.165 [b] | 0.116 [a] | 0.104 [a] | 0.227 [c] | 0.105 [a] |
| 3 | 0.141 [ab] | 0.122 [ab] | 0.173 [bc] | 0.207 [c] | 0.106 [a] | 0.109 [a] |
| 4 | 0.157 [a] | 0.162 [a] | 0.143 [a] | 0.272 [b] | 0.153 [a] | 0.118 [a] |
| 5 | 0.176 [ab] | 0.215 [a] | 0.212 [a] | 0.316 [c] | 0.134 [b] | 0.178 [ab] |
| 6 | 0.247 [a] | 0.246 [a] | 0.257 [a] | 0.331 [d] | 0.182 [c] | 0.089 [b] |
| 7 | 0.127 [a] | 0.156 [c] | 0.107 [ab] | 0.129 [a] | 0.234 [d] | 0.097 [b] |
| 8 | 0.170 [a] | 0.187 [a] | 0.309 [b] | 0.275 [b] | 0.197 [a] | 0.156 [a] |
| 9 | 0.207 [a] | 0.207 [a] | 0.212 [a] | 0.234 [ab] | 0.225 [ab] | 0.241 [b] |
| 10 | 0.204 [a] | 0.226 [ab] | 0.283 [bc] | 0.304 [c] | 0.236 [abc] | 0.181 [a] |
| relative $C_{HA}$ [1] | 0.915 [ab] | 1.010 [a] | 1.053 [ac] | 1.254 [c] | 1.002 [a] | 0.766 [b] |

Values within the row marked with the same letters are not different at the $p < 0.05$ level of significance (Fisher's test). [1] relative to the average value of a variable within each experimental site.

**Table 5.** The $C_{HA}/C_{FA}$ and E4/E6 ratio in soil at the individual experimental sites.

| Site/TRT | Unfert. | NPK | FYM | FYM + NPK | STRAW/BT | STRAW/BT + NPK |
|---|---|---|---|---|---|---|
| $C_{HA}/C_{FA}$ | | | | | | |
| 1 | 0.783 [a] | 0.655 [a] | 0.804 [a] | 0.828 [a] | 0.628 [a] | 0.613 [a] |
| 2 | 0.545 [a] | 0.924 [b] | 0.732 [ab] | 0.664 [a] | 0.941 [b] | 0.634 [a] |
| 3 | 0.495 [a] | 0.616 [ab] | 0.814 [ab] | 0.882 [b] | 0.737 [ab] | 0.546 [ab] |
| 4 | 0.552 [a] | 0.694 [ab] | 0.655 [ab] | 0.900 [b] | 0.566 [a] | 0.480 [a] |
| 5 | 0.743 [ab] | 0.880 [b] | 0.879 [b] | 0.833 [ab] | 0.584 [a] | 0.578 [a] |
| 6 | 0.648 [d] | 1.223 [a] | 1.351 [ab] | 1.390 [ab] | 1.666 [bc] | 1.908 [c] |
| 7 | 0.640 [b] | 1.007 [c] | 0.489 [a] | 0.648 [b] | 0.976 [c] | 0.479 [a] |
| 8 | 0.724 [a] | 0.706 [a] | 1.381 [b] | 1.026 [ab] | 0.708 [a] | 0.672 [a] |
| 9 | 1.062 [ab] | 1.820 [c] | 1.375 [ac] | 1.374 [ac] | 0.900 [b] | 1.008 [ab] |
| 10 | 0.801 [a] | 0.993 [ab] | 1.340 [b] | 1.278 [b] | 0.882 [a] | 0.719 [a] |
| relative $C_{HA}/C_{FA}$ [1] | 0.826 [b] | 1.089 [a] | 1.120 [a] | 1.134 [a] | 0.990 [ab] | 0.840 [b] |
| E4/E6 | | | | | | |
| 1 | 6.26 [a] | 6.12 [a] | 5.86 [a] | 6.05 [a] | 6.14 [a] | 6.24 [a] |
| 2 | 5.52 [c] | 5.87 [a] | 5.97 [a] | 6.18 [a] | 6.77 [b] | 6.97 [b] |
| 3 | 5.58 [c] | 5.66 [c] | 6.31 [a] | 6.56 [ab] | 6.62 [ab] | 7.00 [b] |
| 4 | 6.12 [bc] | 5.88 [ab] | 6.68 [d] | 5.72 [a] | 6.02 [abc] | 6.31 [cd] |
| 5 | 6.91 [ab] | 6.95 [ab] | 7.08 [b] | 6.51 [a] | 3.57 [c] | 3.91 [c] |
| 6 | 4.08 [a] | 5.08 [b] | 4.78 [b] | 4.28 [a] | 4.10 [a] | 4.22 [a] |
| 7 | 5.04 [a] | 5.07 [a] | 5.08 [a] | 5.36 [a] | 5.12 [a] | 5.35 [a] |
| 8 | 5.39 [ac] | 4.41 [c] | 5.80 [ab] | 6.30 [ab] | 6.07 [ab] | 6.83 [b] |
| 9 | 4.98 [d] | 4.35 [c] | 4.19 [bc] | 3.94 [b] | 3.50 [a] | 3.24 [a] |
| 10 | 6.60 [c] | 5.58 [e] | 6.52 [bc] | 6.28 [ab] | 3.72 [d] | 6.06 [a] |
| relative E4/E6 [1] | 1.019 [ab] | 0.993 [ab] | 1.046 [b] | 1.023 [ab] | 0.922 [a] | 0.998 [ab] |

Values within the row marked with the same letters are not different at the $p < 0.05$ level of significance (Fisher´s test). $C_{HA}/C_{FA}$—humic to fulvic acid carbon ratio, E4/E6—absorbances ratio at the wavelengths of 465 and 665 nm. [1] relative to the average value of a variable within each experimental site.

**Table 6.** Humification rate (HR) and humification index (HI) in soil at the individual experimental sites.

| Site/TRT | Unfert. | NPK | FYM | FYM + NPK | STRAW/BT | STRAW/BT + NPK |
|---|---|---|---|---|---|---|
| HR | | | | | | |
| 1 | 0.246 [b] | 0.339 [a] | 0.297 [a] | 0.313 [a] | 0.222 [b] | 0.298 [a] |
| 2 | 0.304 [b] | 0.255 [ab] | 0.243 [ab] | 0.218 [a] | 0.201 [a] | 0.213 [a] |
| 3 | 0.394 [ab] | 0.322 [abc] | 0.344 [ab] | 0.412 [b] | 0.233 [c] | 0.285 [ac] |
| 4 | 0.380 [c] | 0.338 [abc] | 0.281 [ab] | 0.364 [bc] | 0.331 [abc] | 0.277 [a] |
| 5 | 0.288 [a] | 0.327 [ab] | 0.339 [ab] | 0.513 [c] | 0.288 [a] | 0.371 [b] |
| 6 | 0.278 [a] | 0.261 [a] | 0.247 [a] | 0.383 [d] | 0.203 [c] | 0.139 [b] |
| 7 | 0.365 [a] | 0.361 [a] | 0.371 [a] | 0.368 [a] | 0.527 [b] | 0.349 [a] |
| 8 | 0.426 [ab] | 0.430 [ab] | 0.475 [ab] | 0.494 [b] | 0.478 [ab] | 0.378 [a] |
| 9 | 0.310 [a] | 0.243 [c] | 0.278 [d] | 0.314 [a] | 0.378 [b] | 0.378 [b] |
| 10 | 0.331 [ab] | 0.336 [ab] | 0.337 [ab] | 0.345 [b] | 0.331 [ab] | 0.280 [a] |

**Table 6.** *Cont.*

| Site/TRT | Unfert. | NPK | FYM | FYM + NPK | STRAW/BT | STRAW/BT + NPK |
|---|---|---|---|---|---|---|
| relative HR [1] | 1.027 ab | 0.990 a | 0.981 a | 1.142 b | 0.957 a | 0.903 a |
| HI | | | | | | |
| 1 | 0.108 ab | 0.134 a | 0.132 a | 0.142 a | 0.086 b | 0.113 ab |
| 2 | 0.107 bc | 0.121 c | 0.103 abc | 0.087 ab | 0.097 ab | 0.082 a |
| 3 | 0.130 ab | 0.123 ab | 0.154 bc | 0.190 c | 0.096 a | 0.100 a |
| 4 | 0.134 ac | 0.136 ac | 0.111 ab | 0.170 c | 0.120 ab | 0.090 b |
| 5 | 0.122 ab | 0.153 a | 0.158 a | 0.233 c | 0.106 b | 0.136 ab |
| 6 | 0.161 a | 0.161 a | 0.162 a | 0.211 d | 0.116 c | 0.055 b |
| 7 | 0.142 a | 0.181 c | 0.120 ab | 0.145 a | 0.260 d | 0.113 b |
| 8 | 0.174 a | 0.177 a | 0.263 c | 0.250 bc | 0.192 ab | 0.148 a |
| 9 | 0.159 ab | 0.155 b | 0.161 ab | 0.181 ac | 0.179 ac | 0.190 c |
| 10 | 0.148 ab | 0.165 a | 0.193 a | 0.194 a | 0.155 ab | 0.117 b |
| relative HI [1] | 0.954 a | 1.041 a | 1.054 a | 1.225 c | 0.946 ab | 0.780 b |

Values within the row marked with the same letters are not different at the $p < 0.05$ level of significance (Fisher´s test). HR = $(C_{FA} + C_{HA})/TOC$, HI = $C_{HA}/TOC$. [1] relative to the average value of a variable within each experimental site.

The FYM + NPK treatment resulted in an increased relative $C_{HA}$ content compared to the unfertilized treatment, NPK treatment, STRAW/BT and STRAW/BT + NPK treatment. The STRAW/BT + NPK treatment led to a decrease in relative $C_{HA}$ content compared to all other treatments except the unfertilized. Even though the relative $C_{HA}$ content did not differ between the FYM treatment and the STRAW/BT treatment, in absolute figures, the FYM treatment achieved higher $C_{HA}$ content compared to the STRAW/BT treatment at half of the experimental sites.

Compared to the unfertilized treatment, the relative $C_{HA}/C_{FA}$ ratio was increased in the NPK, FYM and FYM + NPK treatments (Table 5). The FYM + NPK treatment resulted in an increased relative $C_{HA}/C_{FA}$ ratio in comparison with the STRAW/BT + NPK treatment. A decrease in the relative $C_{HA}/C_{FA}$ ratio was recorded in the STRAW/BT + NPK treatment compared to the NPK treatment. Even though the relative $C_{HA}/C_{FA}$ ratio did not differ between the FYM and the STRAW/BT treatment, in absolute figures, a higher $C_{HA}/C_{FA}$ ratio in the FYM treatment compared to the STRAW/BT treatment was recorded at four experimental sites.

The highest E4/E6 ratio was found in the FYM treatment at half of the experimental sites. Lower values of the relative E4/E6 ratio were recorded in the STRAW/BT treatment in comparison with the FYM treatment. However, no significant difference in relative E4/E6 ratio between the FYM + NPK and STRAW/BT + NPK treatment was recorded.

### 3.2. Degree of Humification

The FYM + NPK treatment resulted in an increased relative humification rate compared to all other treatments except for the unfertilized one (Table 6).

In terms of the relative humification index, the FYM + NPK treatment achieved higher values in comparison with all other treatments while the STRAW/BT + NPK treatment resulted in a lower relative humification index compared to all treatments except for the STRAW/BT one. Even though no significant difference in relative humification index between the FYM and the STRAW/BT treatment was found, in absolute numbers, a higher humification index in the FYM treatment compared to the STRAW/BT treatment was recorded at half of the experimental sites.

### 3.3. Carbon Sequestration

The influence of treatment was recorded on neither the relative contribution of organic fertilizers to soil organic carbon stock (RC) nor carbon sequestration efficiency (CSE) at only two experimental sites (Table 7). Unlike the straw return, farmyard manure application

resulted in higher (positive) values of both RC and CSE. On average, 30.75% and 43.20% of the carbon input in farmyard manure was converted to the organic carbon content of the soil in the FYM and the FYM + NPK treatment, respectively.

**Table 7.** The relative contribution of organic fertilizers to soil organic carbon stock (RC) and carbon sequestration efficiency (CSE) at the individual experimental sites.

| Site/TRT | FYM | FYM + NPK | STRAW/BT | STRAW/BT + NPK |
|---|---|---|---|---|
| RC (%) | | | | |
| 1 | 6.27 [a] | 8.28 [a] | 14.96 [a] | 8.90 [a] |
| 2 | 11.89 [b] | 15.42 [b] | −3.25 [a] | 3.25 [a] |
| 3 | −1.88 [ab] | 15.16 [b] | −19.85 [a] | −3.77 [ab] |
| 4 | 16.95 [b] | 10.36 [b] | −4.46 [a] | −7.90 [a] |
| 5 | 10.09 [a] | 7.54 [a] | −2.26 [a] | 8.57 [a] |
| 6 | 8.70 [b] | 8.61 [b] | −12.34 [a] | −11.90 [a] |
| 7 | 8.19 [b] | 4.03 [ab] | −7.45 [a] | −9.45 [a] |
| 8 | 20.68 [b] | 20.82 [b] | −4.93 [a] | −3.16 [a] |
| 9 | 1.04 [b] | −0.62 [b] | −7.62 [a] | −9.16 [a] |
| 10 | 18.83 [ab] | 36.72 [b] | 2.00 [a] | 11.55 [a] |
| CSE (%) | | | | |
| 1 | 16.0 [a] | 22.5 [a] | 36.4 [a] | 21.9 [a] |
| 2 | 21.6 [b] | 28.7 [b] | −5.7 [a] | 5.9 [a] |
| 3 | 44.5 [b] | 29.7 [b] | −11.3 [a] | −21.9 [a] |
| 4 | −12.1 [ab] | 55.1 [b] | −85.9 [a] | −16.3 [ab] |
| 5 | 32.6 [a] | 25.2 [a] | −9.1 [a] | 27.5 [a] |
| 6 | 31.3 [b] | 32.2 [b] | −31.5 [a] | −31.1 [a] |
| 7 | 21.0 [b] | 9.4 [ab] | −18.4 [a] | −24.9 [a] |
| 8 | 63.2 [b] | 67.6 [b] | −14.8 [a] | −9.5 [a] |
| 9 | 3.3 [c] | −2.1 [bc] | −18.1 [ab] | −22.4 [a] |
| 10 | 86.1 [ab] | 163.6 [b] | 8.1 [a] | 47.6 [a] |

Values within the row marked with the same letters are not different at the $p < 0.05$ level of significance (Fisher´s test). RC = [(TOC$_{treatment}$−TOC$_{control}$)/TOC$_{control}$] × 100, CSE = [(TOC$_{treatment}$−TOC$_{control}$)/TCI] × 100.

## 4. Discussion

### 4.1. Fractions of Organic Carbon in Soil

The findings of Kutova et al. [32], who state that mineral fertilization increased the $C_{FA}$ content in soil compared to organic fertilization, were not approved in our research. However, Hao et al. [33] found no difference between the mineral fertilization and straw return with the addition of mineral fertilizer in the $C_{FA}$ content after 13 years of experiments which is in accordance with our results. Unlike Zheng et al. [34] who compared mineral fertilization with deep incorporation of maize straw, no significant difference in the $C_{HA}$ content was recorded between the NPK and the STRAW/BT treatment in our results. The findings of Hao et al. [35] recording decreased $C_{HA}$ content in mineral treatment compared to the straw return with the addition of mineral fertilizers, were not confirmed either.

Although humic acids with larger molecules increased in all manured plots, differences between humic acids in plots with and without manure applied at practical levels in elemental and spectroscopic analyzes were small or scarce [35]. The effect of not only the farmyard manure application but also mineral fertilization, on the $C_{HA}$ content, can be concluded.

In contrast to the findings of Song et al. [28] and Sarma and Gogoi [36], the E4/E6 ratio significantly correlated with neither the $C_{HA}$ content nor the $C_{FA}$ content. Furthermore, Gerzabek et al. [37] and Oktaba et al. [38] recorded a significant effect of different fertilizers on the E4/E6 ratio, while no effect on the $C_{HA}$/$C_{FA}$ ratio was found by the authors. Balik et al. [39] whose research was carried out under similar soil-climatic conditions also state that the E4/E6 ratio did not provide relevant information about soil organic matter quality [39].

Compared to mineral fertilization, straw return [40] and poultry manure [41] increased both the $C_{HA}$ content and the $C_{HA}/C_{FA}$ ratio in soil. This phenomenon was not confirmed in our research, not even in the case of the FYM + NPK treatment, because this treatment led to an increase in both the $C_{HA}$ content and the $C_{FA}$ content.

A higher $C_{HA}/C_{FA}$ ratio and lower E4/E6 ratio were recorded at the experimental sites with lower annual precipitation, which agrees with the results of Larionova et al. [42] and Radmanovic et al. [43].

### 4.2. Degree of Humification

Some studies have shown a decrease in the degree of soil humification as a result of the application of farmyard manure [37,44]. On the other hand, Marinari et al. [45] recorded a higher humification index and a lower proportion of aliphatic and aromatic fractions in soil due to farmyard manure application compared to mineral nitrogen fertilization. The reason for this may be the formation of stable humic substances during the ripening or composting of manure [5]. Wei et al. [46] concluded that long-term fertilization with organic matter with or without NPK could increase the humification degree of soil. However, this phenomenon was shown only in the case of the FYM + NPK treatment in our results. Due to the higher humification rate and humification index in the FYM + NPK treatment compared to the others, according to Tavares and Nahas [6], only this treatment (FYM + NPK) affected microbial composition and activity.

### 4.3. Carbon Sequestration

Even though Ghafoor et al. [47] state that nitrogen fertilization causes greater stabilization of plant residues, presumably due to increased microbial carbon use efficiency, no significant decrease in STRAW/BT compared to the STRAW/BT + NPK one was recorded regarding all studied variables. Furthermore, a negative correlation was found between carbon sequestration efficiency and the weighted average of the C/N ratio in applied fertilizers. According to Wang et al. [31], carbon sequestration efficiency is primarily related to soil fertility. This can explain a positive correlation of the RC and carbon sequestration efficiency with the $C_{HA}$ content and a negative correlation with carbon input and the C/N ratio in applied fertilizers because, according to Klik et al. [48], a higher content of stable carbon forms ($C_{HA}$) is beneficial for carbon sequestration in soil. Organic inputs to soil with a high C/N ratio lose more carbon in turnover than the amendments with a low C/N ratio [49]. According to Wang et al. [31], the effect of straw return on soil organic carbon stock is attributed to site-specific conditions; straw return did not significantly increase soil organic carbon stocks at the experimental site with low soil organic carbon density (13.5 g/kg), while the carbon pool was enhanced at the sites with high soil organic carbon contents (24.5 and 31.3 g/kg). However, our research was conducted on soils corresponding to a low organic carbon content (12.5 g/kg on average), which can clarify no positive effect of straw incorporation on the soil organic carbon stock.

A significant correlation between humification index and both the RC and carbon sequestration efficiency is in accordance with the findings of Mockeviciene et al. [50] and Hao et al. [33] who state that polymerization of humic acids, i.e., higher humification index, creates more favorable conditions for carbon sequestration.

### 4.4. Treatment

Although carbon input did not significantly correlate with any soil organic carbon fractions, the C/N ratio of applied organic fertilizers significantly correlated with the $C_{HA}$ content and humification index (moderate correlation). Balik et al. [50] confirmed in another experiment that the $C_{HA}$ content and the $C_{HA}/C_{FA}$ ratio were affected by the C/N ratio in applied fertilizer, while the effect of the amount of carbon input was not recorded or only led to an increase in the $C_{FA}$ content [51,52]. The initial C/N ratio of the substrate has a significant effect on the microbial community and degradation of organic matter [53]. A decrease in the C/N ratio occurs during composting and the C/N ratio is a common

indicator of compost maturity [54]. The initial C/N ratio of 25 favors the formation of high-quality compost [54], which is closer to the C/N ratio of manured treatments (C/N = 38) compared to the treatments with straw return (C/N ratio > 60).

The STRAW/BT + NPK treatment brought no benefit in comparison with the NPK treatment in terms of soil organic matter quality. The NPK treatment resulted in increased $C_{HA}$ content, $C_{HA}/C_{FA}$ ratio and humification index compared to the STRAW/BT + NPK treatment. Therefore, the results of a 15-year experiment by Hao et al. [55] were supported because the authors stated that the organic matter of the soil under straw return conditions becomes enriched by aliphatic components and reduced by aromatic components, suggesting that the degree of humification of the organic matter of soil decreases with straw return. Similarly, Arlauskiene et al. [56] found a lower $C_{HA}/C_{FA}$ ratio and humification degree after barley straw incorporation into soil compared to the treatment with straw removed from soil. However, the results of Koishi et al. [44] were not supported; these authors stated that in the absence of organic matter input the application of mineral fertilizers alone resulted in decreased soil organic carbon content and an increased humification index.

The FYM treatment resulted only in an increased E4/E6 ratio compared to the STRAW/BT treatment, but despite the findings of Aparna et al. [57], the humification index did not differ among these two treatments. Except for the TOC content, no soil organic carbon fractions or humification indices differed between the FYM and the NPK treatments. A significant increase in the $C_{HA}$ content, $C_{HA}/C_{FA}$ ratio, humification rate and humification index was recorded in the FYM + NPK treatment in comparison with the STRAW/BT + NPK treatment. The STRAW/BT + NPK treatment brought no benefits in comparison with the FYM + NPK treatment. The FYM + NPK treatment led to a higher content of $C_{FA}$ and $C_{HA}$, a humification rate and a humification index compared to the NPK treatment.

## 5. Conclusions

A significant correlation between the E4/E6 ratio and soil organic carbon fractions, humification rate, and humification index was not recorded. Additionally, no significant correlation was found between the carbon input applied in fertilizers and the organic carbon fractions of the soil. On the other hand, the weighted average of the C/N ratio in organic fertilizers negatively correlated with the humic acid carbon, humification index, and $C_{HA}/C_{FA}$ ratio. Although straw return led to a lower E4/E6 ratio compared to the farmyard manure application, the $C_{HA}/C_{FA}$ ratio was unchanged among these treatments. Only the combination of farmyard manure with mineral NPK resulted in a higher humification index, humification rate, humic acid carbon content and fulvic acid carbon content compared to the application of mineral fertilizers alone. Neither straw return nor the combination of straw return with mineral NPK brought any benefit compared to the application of mineral fertilizers alone in terms of soil organic matter quality. The application of farmyard manure with and without the addition of NPK led to higher values of carbon sequestration efficiency in soil compared to the straw return with and without the addition of NPK. Carbon sequestration efficiency negatively correlated with the weighted average of the C/N ratio in applied fertilizers.

**Author Contributions:** Conceptualization, O.S.; methodology, M.S. and J.B.; validation, J.Č., J.B. and M.K.; formal analysis, O.S.; investigation, O.S. and M.S.; resources, J.B.; data curation, M.K.; writing—original draft preparation, O.S.; writing—review and editing, J.B.; supervision, J.B. All authors have read and agreed to the published version of the manuscript.

**Funding:** This research was supported by the NAZV QK21010124 research project "Soil organic matter–evaluating of quality parameters" financed by the Czech Ministry of Agriculture.

**Data Availability Statement:** Data available from corresponding author.

**Acknowledgments:** The authors would like to thank Petr Kolbábek for the precise chemical analysis.

**Conflicts of Interest:** The authors declare no conflict of interest. The funders had no role in the design of the study; in the collection, analyses, or interpretation of data; in the writing of the manuscript; or in the decision to publish the results.

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
