# Peer review of "Long-Term Application of Organic Fertilizers in Relation to Soil Organic Matter Quality"

_agronomy, doi:10.3390/agronomy13010175_

Round 1

Reviewer 1 Report

Line 76: Table 1 – soil group names are obviously written according to the WRB 2006 (Albeluvisols). The WRB 2015 should be applied. Principal qualifiers – capital letters (Table 1 and 3, text).

Line 267: the CHA/CFA treatment – ratio?

One of the prominent results of this work is summarized in the conclusion: “Unlike the straw return with or without the addition of mineral NPK, farmyard manure application with or without the addition of mineral NPK resulted in positive values of carbon sequestration efficiency”. It should be discussed in the wider context of current animal husbandry; point out its advantages in relation to the other possible environmental effects (i.e. livestock production of greenhouse gases, etc).

Author Response

Dear Madam/Sir,

Thank you for your review. All your comments were implemented into the paper.

Reviewer 2 Report

agronomy-2120474

This study mainly explores the long-term application of organic fertilizers in relation to soil organic matter quality. The experimental design is reasonable. However, there was no in-depth analysis of experimental data. In particular, there is a lack of indicators of organic matter relevant to the topic. Therefore, I suggest that this article can be published after major revision.

1.     Line 23: The research parameters should specify which indicators are included.

2.     The abstract lacks a concluding sentence and needs to be supplemented.

3.     The introduction should correspond to the topic, focusing on organic matter rather than focusing on organic carbon and humic acids.

4.     Please divide the result part into subheadings. It is better to use different subheadings to distinguish different categories of indicators.

5.     In my opinion, the content of the results is not rich enough. The results of one paper supported by three tables are not convincing. In addition, in order to better fit the title of this article, please add relevant data on organic matter content.

6.     The discussion section needs to be rewritten. The discussion is illogical. It is suggested to discuss the addition of subheadings.

7.     The conclusion part needs to be summarized, to write important conclusions, new findings of this study, and innovative conclusions in the field.

      8. Chart form is too single, recommended figures and tables reasonable allocation.

Author Response

Dear Madam/Sir,

Thank you for your review. All your comments were implemented into the paper. The Introduction, Results and Discussion chapters were rewritten. The relative values are now accompanied by the absolute ones. Both the abstract and conclusions were partly modified.

Reviewer 3 Report

The manuscript has high potential, because it is a study of many years and has several soils, but as the data were analyzed and as the article is written, it needs to be re-analyzed and written.

Some points are highlighted below:

- The introction is poor and could be improved.

- Based on the Agronomy's template, a paragraph in the introduction related to main results and relevance of the manuscript  is missing.

- The authors present some empirical groupings related to soil types, however, without statistical or literature support. This adopted approach compromises the quality of the dataset presented. They assume that the behavior of the same soil class is similar regardless of location, which does not happen in practice, an effect commonly observed in the literature by several works.

- In tables 3 and 5, only the averages are presented, thus, no type of data dispersion information is presented (standard deviation or variance); how did the authors use Anova, how do they ensure that the ANOVA assumptions were met? They need to show how they evaluated all the assumptions.

-What seems to be the main objective of the work is to show the effect of the application of organic fertilizers, but they focus more on the soil effect, the authors should give more emphasis on the studied treatments.

-The whole approach to the results needs to be redone, as the data from the studied treatments were grouped regardless of the study site, and this is a big mistake, as it does not consider that the treatments can have different effects depending on the characteristics of the site and type of soil.

-The literature already shows this soil effect dependent on the application of mineral and organic fertilizers.

- The data analysis and results section needs to be redone.

- The discussion is poor and could be improved.

- According to the data approach in the results section, the discussion should be redone

Author Response

Dear Madam/Sir,

Thank you for your review. All your comments were implemented into the paper. The Introduction, Results and Discussion chapters were rewritten. The relative values are now accompanied by the absolute ones. Both the abstract and conclusions were partly modified. Soil type grouping was omitted. However standard deviations were not added because the affected tables would be very large and probably not easy to read.

Round 2

Reviewer 2 Report

No

Reviewer 3 Report

Dear Editor and Authors.
Highlighted points in the old version have been adjusted and in this new version of the manuscript. The manuscript was improved by separating comparisons by study sites. After the changes made by the authors, the manuscript was improved and could be published in Agronomy Journal in the current form.